# The Development and Validation of the Adolescent Problematic Gaming Scale (PGS-Adolescent)

**DOI:** 10.3390/bs15010013

**Published:** 2024-12-27

**Authors:** Zhanni Luo, Jiayan Xie

**Affiliations:** School of Foreign Languages and Literatures, Chongqing Normal University, Chongqing 401331, China

**Keywords:** video games, problematic gaming, problematic game-playing, gaming addiction, adolescents, scale development

## Abstract

This study aims to develop and validate the Adolescent Problematic Gaming Scale (PGS-Adolescent). Following established scientific protocols, we developed an initial version of the PGS-Adolescent scale and validated it using data from 448 valid survey responses collected from adolescents in China. The dataset was split into two parts: 225 responses were allocated for exploratory factor analysis (EFA), and 223 for confirmatory factor analysis (CFA). The EFA and CFA processes necessitated the removal of 10 items due to low factor loadings, low communalities, misalignment with intended factors, and inadequate item retention. Significantly, all the survey items measuring daily-life disturbance (DD) were excluded. The final 20-item PGS-Adolescent scale includes four constructs: interpersonal relationships (IRE), schooling disruption (SD), physical consequences (Phy-C), and psychological consequences (Psy-C). Researchers may consider directly applying the validated PGS-Adolescent scale or evaluating its applicability and validity in diverse populations and contexts.

## 1. Introduction

### 1.1. Background

As innovative technologies continue to advance and urbanization rapidly reduces safe recreational spaces in cities, traditional games have given way to video games ([16]). Compared to traditional games, video games are more advantageous in integrating various captivating game elements, such as points, badges, leaderboards, sound effects, and game avatars ([35]; [41]; [32]). Video games are also more advantageous in integrating engaging game mechanisms, such as epic meaning, social influences, real-time feedback, challenges, and alternative success ([15]; [32]). The intricately designed game elements and mechanisms greatly enhance the allure of video games ([30]).

Due to the highly captivating nature of video games, players, especially adolescent players, are easily addicted. Researchers indicate that in complex video games, such as massively multiplayer online role-playing games (MMORPGs), there are extensive reward systems that encourage continuous investment of time, energy, and money, ultimately leading to gaming addiction ([41]; [1]). Additionally, due to a higher acceptance of innovations, adolescents are more inclined to become addicted to video games than older players ([16]; [41]; [24]; [23]; [49]). Therefore, for the last few decades, gaming addiction has been recognized as a public health problem ([16]; [49]; [8]; [55]).

### 1.2. Research Gaps

Previous research has identified several gaps. Firstly, the majority of current studies concentrate on gaming addiction, giving limited attention to problematic gaming ([16]). Both gaming addiction and problematic gaming involve behaviors that negatively impacts an individual’s life. However, problematic gaming is typically less severe, whereas gaming addiction is more severe and may even involve pathological issues ([47]; [51]). It can be said that problematic gaming may develop into gaming addiction if appropriate interventions are not provided. The two concepts, gaming addiction and problematic gaming, are not the same ([38]). Failing to distinguish between these concepts could result in the application of diagnostic tools for gaming disorder when evaluating problematic gaming behaviors, leading to inaccurate diagnoses and ultimately impacting interventions for problematic gaming behaviors ([16]; [49]).

Secondly, there is a scarcity of measurement scales specifically designed for assessing problematic gaming among adolescents. Although [47] ([47]) attempted to measure problematic video game playing, and [39] ([39]) evaluated the psychometric properties of a problematic online gaming scale among a nationwide sample of adolescents, the availability of comprehensive measurement tools tailored for adolescent problematic gaming remains insufficient ([19]).

### 1.3. Aims and Significance

This study aims to develop and validate a scale measuring problematic gaming in adolescents, which is named the Adolescent Problematic Gaming Scale (PGS-Adolescent). The PGS-Adolescent scale intends to provide a comprehensive and reliable measurement tool, considering the distinct behavioral patterns and psychological characteristics of adolescents engaged in video game playing activities (see details in Appendix A).

The significance of this research lies in its potential to offer a standardized instrument that enables a more accurate assessment of problematic gaming behaviors in adolescent populations, contributing to both academic understanding and practical interventions in adolescent mental health.

## 2. Literature Review and Scale Development

### 2.1. Definitions of Gaming Addiction and Problematic Gaming

The concept of addiction originates from the medical field, referring to a dependency associated with the intake of substances such as drugs or alcohol ([1]). Addiction is often accompanied by tolerance (the need for increasing amounts of a substance or behavior to achieve the desired effect), withdrawal symptoms (physical and psychological effects that occur when a person stops using a substance or engaging in a behavior to which they are addicted), dependence (the reliance on a substance or behavior to function normally), and social problems (such as interpersonal relationship crises, social isolation, violence, criminal tendencies, etc.) ([22]; [4]; [26]).

The concept of addiction has evolved to encompass various subcategories, including cyber addiction, Internet addiction, gaming addiction, video game addiction, mobile phone addiction, social media addiction, and more ([29]; [46]; [18]; [56]). Among these, video game addiction involves human-computer interactions, falls under the category of behavioral addiction, and is also considered a form of technological addiction ([1]; [25]). As modern technology progresses, video games have become ubiquitous; hence, although games do not inherently necessitate digital technology, the term “gaming addiction” is predominantly used to refer to video gaming addiction ([28]).

Gaming addiction refers to an excessive obsession with games despite negative consequences, which may involve clinically significant impairments in multiple aspects of a person’s life ([24]; [11]). Individuals afflicted with this addiction demonstrate an inability to regulate their excessive gaming behavior, leading to a loss of control ([24]; [20]). Gaming addiction is commonly associated with symptoms such as salience, tolerance, withdrawal, mood modification, relapse, conflict, and other problems ([11]).

A term closely related to gaming addiction is problematic gaming. Problematic gaming refers to a pattern of excessive gaming behavior that can have negative effects on an individual’s life, work, or academic performance ([20]; [2]). However, it does not meet the clinical criteria to be classified as a diagnosable disorder or addiction. This term is often seen as an early or less severe stage in the spectrum of gaming-related issues. It is closely linked or associated with compulsive gaming ([2]). In many instances, the “gaming” in problematic gaming refers to video game playing, digital gaming, or online gaming.

Other concepts closely related to these two (gaming addiction and problematic game playing) are excessive gaming and gaming disorder. Excessive gaming, also known as video game dependency, refers to spending an inordinate amount of time on gaming, surpassing what is considered typical or healthy ([19]). Excessive gaming often does not reach the level of a clinically diagnosed addiction ([10]). A more severe manifestation of this is gaming disorder, which is defined as the persistent and recurrent use of games, leading to clinically significant impairment or distress ([23]; [9]; [7]).

Research on gaming addiction has presented some prominent topics and trends. [21] ([21]) conducted a systematic review of empirical studies on Internet gaming addiction, reporting trends in research on this topic. [21] ([21]) conducted identified three major trends in studies related to Internet gaming addiction. Firstly, there was a focus on etiological risks in some studies, which investigate factors such as personality traits, motivations for gaming, and the structural characteristics of games ([19]). Secondly, some researchers have directed their attention to pathological addiction, investigating the classification, assessment, epidemiology, and phenomenology of Internet gaming addiction. Thirdly, there is a growing interest in exploring the ramifications and consequences of gaming addiction, especially negative outcomes and treatment approaches ([21]). These three major trends are interconnected, as etiological risks can lead to pathological addiction, which in turn may strengthen the former. Similarly, pathological addiction can lead to clinically significant negative consequences for an individual, which may worsen the pathological state, necessitating the pursuit of professional treatment ([21]).

Given the high correlation between problematic gaming and gaming addiction, along with the relatively less attention given to the former and its heightened value for addiction interventions, we have shifted our focus to problematic gaming (also known as problematic video game playing).

Furthermore, as discussed earlier, adolescents are more vulnerable to the negative effects of gaming compared to adults ([24]), prompting our focus on adolescents. Therefore, the primary objective of this study is to develop and validate a specialized scale designed exclusively to assess problematic gaming in adolescents.

### 2.2. Key Constructs in Previous Related Scales

In constructing a scale to measure problematic gaming in adolescents, it is essential to identify the constructs that will comprise the scale ([3]). Constructs, in this context, refer to the theoretical dimensions, concepts, or underlying factors that are believed to represent or explain the phenomenon of interest. 

Our strategy is to integrate the key constructs from existing scales that evaluate equivalent or comparable phenomena, including video game dependency, problematic gaming, gaming addiction, and gaming disorder. Additionally, with the increasing use of smartphones for gaming due to their portability, we also consider scales related to mobile phone gaming addiction, such as the Smartphone Addiction Scale (SAS) ([22]) and the Mobile Phone Problem Use Scale (MPPUS) ([1]). Ultimately, we involved nine related scales, as detailed in Table 1.

We listed the constructs from the nine scales in Table 1, removed those not applicable to our research context, merged similar ones, and obtained nine main constructs that can be used for the PGS-Adolescent scale. These constructs are tolerance, withdrawal, salience, conflict, escape, daily-life disturbance, health, schooling disruption, and problems (see Table 2).

Tolerance measures the extent to which one requires increasingly more time to play games to feel satisfied ([19]; [22]). An example item used to assess tolerance is “I need to spend increasing amounts of time engaged in playing games” ([15]). Concepts closely related to this one includes overuse, loss of control, persistence, and relapse, as they all involve prolonged gaming time and difficulty controlling the time spent on gaming ([23]; [19]; [22]; [2]). A closely related concept is overuse. In the Mobile Phone Dependence Questionnaire scale, [5] ([5]) named one construct “Overuse And Tolerance”, indicating a smartphone-addicted user may try to control phone use but fail to do so, ultimately spending more time on the smartphone to achieve the same level of satisfaction as previously experienced.

Withdrawal encompasses the unpleasant emotional states and physical symptoms that arise when the activity is suddenly stopped or significantly reduced ([24]; [19]; [10]). These effects can manifest psychologically, such as severe mood swings and irritability, or physiologically, including symptoms such as nausea, sweating, headaches, insomnia, and other stress-related reactions ([10]). An item used to assess withdrawal is “If I don’t play for quite a while, I become restless and nervous” ([41]). Similar expressions related to withdrawal include withdrawal symptoms, craving and withdrawal, and withdrawal and escape ([41]; [1]; [25]; [5]).

Salience refers to the phenomenon where a specific activity, in this case gaming, becomes the preeminent aspect of a person’s life, overshadowing other aspects of their mental, emotional, and behavioral landscape ([10]). Salience manifests in thinking as preoccupations and cognitive distortions, in feelings as cravings, and in behavior as deterioration of social behavior and excessive use ([24]; [10]). An example item assessing salience is “My thoughts continually circle around playing video games, even when I’m not playing” ([41]).

Conflict as a construct in related scales mainly refers to interpersonal conflicts. [2] ([2]) explicitly define conflicts as “interpersonal conflicts resulting from excessive gaming”, which exist between the player and those around him or her (p. 5). The main cause of conflict is that players show a preference for excessive gaming over social activities, leading to damage to social relationships. [5] ([5]) add that players might actively invite this outcome because they feel that interacting with the virtual world is more enjoyable than communicating with real-life friends or family. Additionally, gaming addiction or problematic gaming often involves deception, which can further harm the player’s interpersonal relationships. Terms related to conflicts include social isolation, interpersonal conflicts, cyberspace-oriented relationships, virtual life relationships, and lies and deception ([47]; [22]; [2]; [5]).

Escape, or escapism, relates to engaging in a behavior to escape from or relieve negative mood states, such as helplessness, guilt, anxiety, or depression ([23]). Among the nine related scales, six incorporate concepts that are closely related to escape. Positive anticipation in the Smartphone Addiction Scale and mood modification in the Gaming Addiction Scale are analogous to the concept of escape, as they both emphasize the regulation of emotions, striving for positive feelings, and avoiding negative ones ([10]; [22]). An example item assessing escape is “When I feel bad, e.g., nervous, sad or angry, or when I have problems, I use the video games more often” ([47]).

Daily-life disturbance includes missing planned work, having difficulty concentrating in class or while working, experiencing lightheadedness or blurred vision, pain in the wrists or the back of the neck, and sleep disturbances ([22]). According to [22] ([22]), digital technology addiction impacts a wide range of everyday activities. [1] ([1]) established the Mobile Phone Problem Use Scale, in which there is a construct named negative life consequences in the areas of social, familial, work, and financial (p. 42).

Health is a multifaceted concept that involves the well-being of both the body and mind. It is a keyword we have carefully derived from three critical constructs identified in prior related scales: physical symptoms reported in the Japanese version of the Smartphone Dependence Scale ([5]), feeling anxious and lost documented in the Mobile Phone Addiction Index ([25]), and disregard for physical Or psychological consequences outlined in the Problem Video Game Playing Scale ([47]). These constructs collectively paint a picture of health that is not limited to the absence of disease but also includes the overall state of physical and mental well-being. The related survey items include “My shoulders are stiff due to excessive smartphone use” ([5]) and “You feel anxious if you have not checked for messages or switched on your mobile phone for some time” ([25]).

Health is a term that we summarize from our analysis, which is grounded in three constructs identified in previous scales: physical symptoms ([5]), feeling anxious and lost ([25]), and disregard for physical Or psychological consequences ([47]). We consider health to encompass both physical health and mental health. This is supported by related survey items such as “My shoulders are stiff due to excessive smartphone use” ([5]) and “You feel anxious if you have not checked for messages or switched on your mobile phone for some time” ([25]).

Schooling disruption refers to the impact of game addiction on academic performance, including productivity loss and disturbance of concentration in class ([25]; [5]). Since our target population is adolescents, we are particularly focused on the impact of game addiction on their academic performance. Three scales address the impact on learning, described as schooling disruption, productivity loss, and disturbance of concentration in class. Among them, family, schooling disruption not only emphasizes the impact of game addiction on academic performance but also highlights its effects on daily life and interpersonal relationships ([47]); productivity loss emphasizes the loss in learning output ([25]); disturbance of concentration in class emphasizes the inability to concentrate during class ([5]).

Problems in the current research scope indicate the continuation of gaming despite being aware of negative consequences ([23]). [24] ([24]) suggest that it primarily concerns displacement issues, intrapsychic conflict, and subjective feelings of loss of control. Consequently, the scope of problems is quite broad and overlaps with other constructs. For instance, in the Gaming Addiction Scale, majority of conflicts also fall under the category of problems. Therefore, we believe that problems is overly broad and should be further refined.

### 2.3. Scale Development

In developing the Adolescent Problematic Gaming Scale (PGS-Adolescent), we have carefully synthesized five key constructs based on those found in previous related scales. These constructs are daily-life disturbance (DD), interpersonal relationship estrangement (IRE), schooling disruption (SD), physical consequences (Phy-C), and psychological consequences (Psy-C) (see Table 3).

Firstly, we excluded the constructs of tolerance and salience due to varied emphases in scale development. Tolerance, which is characterized by the escalating need to engage in gaming for longer durations to attain a sense of satisfaction, is a common element across the nine related scales examined in our study. Our scale prioritizes the measurement of problematic video gaming by focusing on its negative impacts. However, tolerance is often gauged through items that highlight the desire to play more, rather than the negative outcomes ([2]). Salience also applies, emphasizing the degree to which gaming activities dominate a person’s life and thoughts; although this does not necessarily accompany negative outcomes (e.g., “spent much free time on games”) ([24]). Therefore, the two constructs were removed.

Secondly, we excluded the “Problems” construct due to its lack of specificity. The term “problems” was utilized in two scales: the Internet Gaming Disorder Scale ([23]) and the Gaming Addiction Scale ([24]). According to the definitions provided by the authors, “problems” refers to the issues that gaming addiction or disorder brings to various aspects of players’ lives, such as estrangement in interpersonal relationships, disruptions in education, and intrapsychic conflicts ([24]). The definition is overly broad, and its concept can be encompassed by other constructs; hence, we have excluded “problems” in developing the PGS-Adolescent scale.

Thirdly, we divided the “Health” concept into two specific categories: physical health and mental health. Different scales have different emphases in measuring health-related aspects. For example, the Mobile Phone Dependence Questionnaire leans towards physical health, containing a construct named physical symptoms ([5]), while the Mobile Phone Addiction Index leans towards mental health, containing a construct named Feeling Anxious and Lost ([25]). In establishing the Problem Video Game Playing Scale, [47] ([47]) specified a construct named disregard for physical or psychological consequences. It is evident that the concept of health in the development of the PGS-Adolescent scale can be interpreted as encompassing physical consequences and psychological consequences. Therefore, in the current study, we replaced the term health with physical consequences and psychological consequences.

Fourthly, we replaced the terms “Withdrawal” and “Escape” with “Psychological Consequences”. Withdrawal refers to the distressing emotions experienced when individuals are abruptly cut off from gaming or significantly curtail their gaming activities ([11]). Although its focus is on the sudden discontinuation or reduction of gaming, we believe that its core manifestation is still the unpleasant feelings, which can be reflected in negative psychological consequences. The survey questions measuring withdrawal typically focus on psychological consequences, such as anger or frustration ([23]), irritability and dissatisfaction ([41]), sadness ([25]), or general distress ([47]). The concept of escape is also somewhat similar, as the survey item measuring it is often related to bad feelings, such as anger, sadness, loneliness, and the sense of isolation ([47]; [25]). [25] ([25]) considers withdrawal and escape as the same concept, listing the two as one construct. Consequently, we have replaced both “Withdrawal” and “Escape” with the more comprehensive term “Psychological Consequences”.

Additionally, we refined the “Conflict” construct to “Interpersonal Relationship Estrangement”, providing a clearer depiction of how conflicts manifest in the context of problematic video game playing. Conflict has frequently appeared as a construct in previous scales, such as the Internet Gaming Disorder Scale ([23]), the Video Game Dependency Scale ([41]), and the Gaming Addiction Scale ([24]). When defining conflict in the context of excessive gaming, [10] ([10]) explicitly states that it refers to interpersonal conflicts caused by excessive gameplay, which occur between the player and the people around them. Deception, a common construct in measuring gaming addiction, also falls under interpersonal conflicts, primarily characterized by players lying to parents or partners about the time they spend playing games. In the Problematic Online Gaming Questionnaire ([2]), one construct is named interpersonal conflicts. We believe this construct is valuable; hence, we have retained it while merely renaming it to interpersonal relationship estrangement to better reflect its nature.

We then revised the term “Schooling Disruption” as “Schooling Disturbance”, reflecting the less severe nature of the impact of problematic gaming compared to gaming addiction. This adjustment acknowledges that while the effects of problematic gaming can be negative, they may not always result in the profound disruptions. In previous scales, related expressions included productivity loss ([25]) and Disturbance of Concentration in Class ([5]).

Finally, the term “Daily-life Disturbance” has been retained as is. Some scholars, when describing daily-life disturbances, have touched on concepts overlapping with other constructs, such as wrist pain ([22]), which should be categorized under the physical consequences construct in the current study. However, we believe that the concept itself still holds value. We considered items that reflect the negative impact of video game playing on daily activities, sleep, and diet, such as “I played video games while walking” (DD3), “I played video games when I should be sleeping” (DD4), and “I was so immersed in video game playing that I forgot to eat” (DD5). [1] ([1]) suggest that researchers should consider the negative financial consequences of problematic technology use. Therefore, we retained the item “I didn’t mind spending money on paid gaming applications”. We also considered the excessive time spent and the loss of concentration in studies.

During the literature review, we also identified other constructs, such as relapse, which refers to the tendency to repeatedly revert to earlier patterns of gameplay ([24]; [10]). However, due to its infrequent occurrence, we did not consider incorporating it into the scale development process.

In summary, this study has identified five constructs related to problematic gaming among adolescents: daily-life disturbance (DD), interpersonal relationship estrangement (IRE), schooling disturbance (SD), physical consequences (Phy-C), and psychological consequences (Psy-C).

Drawing on the phrasing of survey items from previous related scales, we have formulated the initial PGS-Adolescent scale, as detailed in Table 3.

## 3. Methods

### 3.1. Research Procedure

This is a scale development study that mainly consists of several key research stages, which follow the guidelines of [3] ([3]), [31] ([31]), and [33] ([33]).

Stage 1 was to confirm the constructs of the scale. Constructs, in scale-development contexts, refer to the theoretical constructs, concepts, or underlying factors that are believed to represent or explain the phenomenon of interest ([3]). Our strategy is to integrate constructs from related scales, remove those that do not fit our study’s scope, adjust the expressions, and finally obtain five constructs: daily-life disturbance (DD), interpersonal relationship estrangement (IRE), schooling disturbance (SD), physical consequences (Phy-C), and psychological consequences (Psy-C) (see Table 2 and Table 3).

Stage 2 involved establishing the item pool. In scale development studies, an item, or survey item, refers to the specific question or statement included in a survey, which is designed to collect data from participants. Survey items are the building blocks of a survey and are designed to elicit responses that can be analyzed to understand attitudes, behaviors, opinions, or other characteristics of the survey’s target population. [3] ([3]) suggest starting with a large set of items when developing a scale, which helps ensure comprehensive coverage. To build a survey item pool, various strategies can be employed, such as adapting survey items from existing scales, drafting survey items based on theories, and transforming qualitative responses into survey items. For this study, we primarily utilized the first approach, focusing on modifying existing survey items to align with the specific research scope of our project (see Table 3).

In Stage 3, we sought expert validation. We provided the drafted PGS-Adolescent scale to two experts and asked them to evaluate its logical consistency and linguistic appropriateness. Both experts hold doctoral degrees, serve as professors, and have published academic research on gaming addiction. They confirmed that the scale’s items encompass the necessary concepts, align with our research objectives and theoretical framework and are expressed clearly ([3]). Additionally, the experts assessed the relevance of each item to the scale’s main theme, its necessity, and its adaptability for diverse cultural backgrounds. Subsequently, we obtained an initial PGS-Adolescent scale consisting of 30 items, which can generally be completed in 5 to 8 min (see Table 3).

Stage 4 was dedicated to data validation. After the PGS-Adolescent scale was reviewed and confirmed by experts, we distributed it to potential participants. Once we collected sufficient survey responses, we excluded those that failed on the reverse question and filter question. Then, utilizing the software SPSS 26.0, we analyzed the mean and standard deviation of each item, assessed the Cronbach’s alpha values for each construct, and evaluated the factor analysis suitability of the entire scale, followed by an exploratory factor analysis (EFA) study. This process was iterative: if an item was removed, we reapplied scale evaluation techniques to ensure the updated scale remained reliable, which could lead to further item removal ([33]). After several rounds of data analysis, we obtained the validated PGS-Adolescent scale, as will be detailed in the Findings section.

Stage 5 was dedicated to evaluating the construct validity and establishing the reliability of the PGS-Adolescent scale as a measurement instrument. We conducted a confirmatory factor analysis (CFA) to test the hypothesized factor structure derived from the EFA results. This involved evaluating the goodness-of-fit indices, such as the Chi-square statistic, RMSEA, and CFI, to confirm whether the data aligned with the proposed model. Additionally, we examined factor loadings to ensure each item appropriately loaded onto its designated construct, reinforcing the scale’s validity. The outcomes of this analysis will be discussed in detail in the Findings section.

### 3.2. Questionnaire Design and Data Collection

Although we have completed the development of the PGS-Adolescent scale through extensive work, as outlined above, it still needs to be converted into a questionnaire before it can be administered. A scale is generally a tool used to assess a particular concept or attribute, whereas a questionnaire is a more comprehensive instrument intended to gather data on a range of topics or constructs. For the questionnaire, we selected a 5-point Likert scale, where 1 represents “strongly disagree” and 5 represents “strongly agree”. Based on the drafted PGS-Adolescent scale, we have added items for demographic data collection (such as age and gender), incorporated a pair of reverse-coded questions, and included a filter question.

The reverse-coded questions were: “I have lied to my parents about the time I spent playing video games” (IRE5) and “I have NOT lied to my parents about the time I spent playing video games.” We hypothesized that participants who carefully read and understood the items would provide opposing responses to these questions, such as “disagree” for one and “agree” for the other. If responses to both items are identical—either both “agree” or both “disagree”—the survey responses are considered unreliable and will be excluded from the analysis.

The filter item instructed participants to select “2—Disagree” on a 5-point Likert scale. Adolescents with problematic gaming behaviors are likely to respond with higher levels of agreement (e.g., “4—Agree” or “5—Strongly Agree”) to most survey items. Participants who fail to read the instructions carefully may answer this item habitually with “4” or “5.” This deliberate choice helps to identify participants who may be hastily agree with statements without fully engaging with the content, thereby ensuring that the data collected more accurately reflects careful consideration and genuine responses ([34]). Any response other than “2—Disagree” was deemed invalid, and such surveys were excluded from the dataset.

We aimed to collect data in China, so all survey items were translated into Chinese. To ensure the accuracy of the translation, we employed the back-translation method. Specifically, this involved translating the survey items from English to Chinese, and then having a separate translator translate them back from Chinese to English. By comparing the original and back-translated versions, we ensured that the meaning and intent of the items were preserved.

During the data collection phase, we used both paper-based and electronic surveys. For young adolescents, we primarily obtained permission from schools to distribute paper questionnaires in classrooms, with the request that participants take them home for their guardians to complete and return. For adolescents aged 18 and above, we mainly shared the link to the electronic questionnaire within university freshman groups and asked them to fill it out. We conducted the survey in four rounds, each lasting approximately 14 days. The data collection started in January 2024 and ended in May, lasting for four months.

### 3.3. Participants and Ethical Considerations

The subjects of this study are adolescents with experience in video game playing. According to the World Health Organization, adolescents are defined as individuals between the ages of 10 and 19, who are experiencing a transitional period between childhood and adulthood ([53]). However, researchers have noted that biological growth and significant social role transitions have evolved over the past century. Therefore, researchers propose an expanded and more inclusive definition of adolescence for the appropriate framing of laws, social policies, and service systems ([43]). Consequently, [43] ([43]) proposed that adolescence encompasses individuals aged 10 to 24 years. We adopt the perspective of [43] ([43]), restricting our participants to adolescents aged between 10 and 24.

We collected survey responses from 594 adolescents in China, of which 146 were excluded for failing to meet the criteria established by the reverse questions and filter item. This resulted in 448 valid responses. The dataset was subsequently divided into two subsets: 225 responses were allocated for exploratory factor analysis (EFA) and 223 for confirmatory factor analysis (CFA). Female participants constituted the majority in both subsets (see Table 4).

Due to the lack of ethics approval to collect additional demographic data, such as age and ethnicity, we were unable to report on these aspects. However, before data collection, we have confirmed with the participants that they were aged between 10 and 24.

In adherence to ethical principles, we thoroughly outlined the study’s objectives and ensured that all participants were fully informed of their rights and interests prior to completing the survey. Participation was strictly voluntary and maintained anonymity throughout. For those under the age of 18, we obtained consent from both the participant and their legal guardian. This study has undergone and successfully passed a rigorous ethical review process.

### 3.4. Data Analysis, Reliability, and Validity

In the data analysis section, we adhered to the guidelines of [3] ([3]), [31] ([31]), and [33] ([33]).

Initially, we examined the internal reliability of the entire initial scale and each construct, ensuring that their Cronbach’s alpha exceeded the benchmark of 0.7. Subsequently, we assessed the Kaiser-Meyer-Olkin (KMO) measure, ensuring its value was above an acceptable threshold, typically 0.6 or higher.

We then conducted an exploratory factor analysis (EFA) using Principal Components Analysis (PCA) with Promax rotation and Kaiser normalization. We ensured that communalities were above 0.5, and that the factor loadings for each survey item exceeded 0.5, with items for the corresponding construct correctly identified within the factor extraction ([31], [33]). For instance, survey items designed to measure daily-life disturbance should have the highest factor loadings on the daily-life disturbance construct and cluster together with other items measuring daily-life disturbance.

Furthermore, we verified the composite reliability (CR) and average variance extracted (AVE) for each construct, ensuring that the former was above 0.7 and the latter above 0.5. This step is crucial for establishing the scale’s convergent validity. We also examined the discriminant validity by checking whether the square root of AVE for each construct was greater than the inter-construct correlations, confirming that the constructs were distinct from each other.

Next, we conducted a confirmatory factor analysis (CFA), with particular attention to the goodness-of-fit indices to verify the applicability and stability of the measurement model. This involved evaluating the goodness-of-fit indices, such as the Chi-square statistic, RMSEA, and CFI.

## 4. Findings

A reliability analysis was conducted on the full dataset of 30 items to evaluate the internal consistency of the scale. The analysis showed a high level of reliability, with a Cronbach’s alpha of 0.962, which notably exceeds the standard threshold of 0.70 ([13]). Further examination of individual constructs confirmed strong internal consistency across the board: daily-life disturbance had a Cronbach’s alpha of 0.862, interpersonal relationship estrangement showed a Cronbach’s alpha of 0.867, schooling disturbance achieved a Cronbach’s alpha of 0.895, physical consequences scored a Cronbach’s alpha of 0.927, and psychological consequences had a Cronbach’s alpha of 0.891.

The Kaiser-Meyer-Olkin (KMO) measure of sampling adequacy reached 0.946, suggesting high suitability for factor analysis. Furthermore, Bartlett’s Test of Sphericity produced a significant result with an approximate chi-square value of 8694.158 (df = 435, *p* < 0.001), affirming the presence of sufficiently large correlations between items for factor analysis to proceed.

Following confirmation of the dataset’s suitability for factor analysis, an exploratory factor analysis (EFA) was conducted utilizing Principal Components Analysis (PCA) with Promax rotation and Kaiser normalization. Initially, items with communalities below 0.50 were removed, including DD6 (0.371), IRE5 (0.427), and Psy-C1 (0.446). Despite efforts to converge, rotation failed after 25 iterations. Consequently, factor extraction was based on Eigenvalues greater than one, resulting in the retention of four components.

Subsequently, survey items with factor loadings below 0.5 were removed: DD2 (0.467), DD4 (0.498), and Phy-C2 (0.493). The daily-life disturbance (DD) construct was then reduced to only three survey items; however, they still failed to converge into a single component. This suggests that the items within this construct possess low internal consistency or measure different underlying structures. Considering this, it was deemed appropriate to remove or redesign these survey items. Consequently, the remaining three DD items were also removed from the analysis.

After removing the six items from the DD construct, we conducted another round of factor analysis on the remaining items. It was observed that SD6 did not load onto the same factor as the other SD items, indicating a lack of convergence. Therefore, SD6 was also removed from the analysis.

In the end, ten items were removed from the initial PGS-Adolescent scale, including six measuring daily-life disturbance, one measuring interpersonal relationship estrangement, one measuring schooling disturbance, one measuring physical consequences, and one measuring psychological consequences (see Table 5).

Table 6 presents the descriptive statistics for each item in the revised PGS-Adolescent scale, along with the results of the exploratory factor analysis. Results showed that the factor analysis extracted four factors that collectively explain 74.34% of the total variance. The Cronbach’s alpha for the entire scale is exceptionally high at 0.953. The Cronbach’s alpha values for each individual construct range from 0.879 to 0.922, all of which significantly exceed the benchmark of 0.7. This surpassing of the standard threshold highlights the strong internal consistency within each construct, suggesting that the items in each construct are closely related and effectively measure the intended constructs.

Moreover, the factor loadings of all the survey items are above the benchmark of 0.5, with some reaching as high as 0.941. In addition, the composite reliability (CR) for each construct exceeds the threshold of 0.7, and the average variance extracted (AVE) surpasses the standard of 0.5. These metrics confirm the high level of internal consistency and convergent validity of the scale. The high AVE values, in particular, suggest that the constructs are well-defined and that the items within each construct are capturing a significant amount of the variance attributed to the constructs.

In conclusion, the analysis presented in Table 6 demonstrates that the revised PGS-Adolescent scale possesses excellent psychometric properties, making it a reliable and valid tool for assessing problematic gaming in adolescents.

The Fornell-Larcker criterion provides guidance for evaluating the discriminant validity of constructs within a measurement model. According to this criterion, the square root of the average variance extracted (AVE) for a construct should exceed the correlations of that construct with other constructs in the model. In this study, the scale meets the criterion, as the square root of AVE for each construct exceeds 0.70, while the correlations between them are all below 0.70 (see Table 7).

The validated PGS-Adolescent scale comprises 20 survey items, organized into four categories: interpersonal relationship estrangement (IRE), schooling disturbance (SD), physical consequences (Phy-c), and psychological consequences (Psy-c).

Next, we conducted a confirmatory factor analysis (CFA). To ensure robustness, we collected an additional 224 valid responses. The results indicated that the chi-square/df ratio was 2.252, falling within the ideal range of 1 to 3. The Incremental Fit Index (IFI) was 0.938, the Tucker-Lewis Index (TLI) was 0.927, and the Comparative Fit Index (CFI) was 0.938, all exceeding the acceptable threshold of 0.90. The Root Mean Square Error of Approximation (RMSEA) was 0.075, which, although above the 0.05 threshold for excellence, remains within the acceptable range of 0.05 to 0.08. The model fit benchmarks were set in alignment with the recommendations provided by [13] ([13]) and [14] ([14]). Collectively, these findings confirm a good fit for the model (see Table 8 and Figure 1).

## 5. Discussion

### 5.1. Discussion on the Removed Survey Items

In this study, ten survey items were removed for various reasons, primarily due to low communalities, low factor loadings, factor misassignment, and insufficient item retention.

Low communalities suggest that an item does not share much variance with other items in the analysis, indicating that it may not adequately represent the underlying construct. In this study, the items “I didn’t mind spending money on video games” (DD6), “I had lied to my parents about the time I spent playing video games” (IRE5), and “I felt bad after playing video games for a long time” (Psy-C1) were removed due to low communalities.

The survey item DD6 was adapted from the Smartphone Addiction Scale by [22] ([22]), originally phrased as “Not minding spending money on paid smartphone applications” ([22]). DD6 was intended to capture the phenomenon where problematic gaming or gaming addiction leads players to uncontrollably indulge in gaming-related impulses, resulting in excessive spending and subsequent daily-life disturbance ([22]). This survey item significantly differs from other items within the same construct, as none of the other items involve money. Moreover, popular commercial games, such as the globally renowned League of Legends, offer enjoyment to players at no cost, suggesting that spending money may not be a reliable measure of problematic gaming among adolescents. Additionally, the phrase “spending money” itself is ambiguous, as it does not specify the amount spent. While excessive spending undoubtedly indicates problematic gaming or even gaming addiction ([22]), spending several dollars to buy a game in the game center may not be related to problematic gaming. Therefore, we consider DD6 unsuitable for inclusion in the PGS-Adolescent scale.

IRE5 is “I had lied to my parents about the time I spent playing video games”, which was adapted from the Internet Gaming Disorder Scale’s “Have you lied to your parents or partner about the time you spent playing games?” ([23]) and the Gaming Addiction Scale’s “Have you lied about time spent on games?” ([24]). We believe that IRE5 was removed because it resembles a lie-detection question, which inherently prompts participants to provide false responses. Lie-detection questions, also known as deception items, are designed to assess the truthfulness of respondents’ answers in surveys or assessments, such as the reversed items in the Eysenck Personality Questionnaire Brief Version ([42]). Typically, lie-detection questions are formulated to resemble statements or scenarios that are difficult for individuals to truthfully affirm or deny. The premise behind these questions lies in the assumption that individuals may feel compelled to lie or provide misleading responses due to social desirability bias, self-presentation concerns, or attempts to conceal sensitive information ([6]). Common lie-detection inquiries involve posing statements such as “I have never lied” or “I always adhere to the rules”, where the socially acceptable or convenient response might be to “agree”, yet the truthful response, acknowledging the universal tendency to deceive or bend the rules at times, would be “disagree”. IRE5 closely resembles the typical lie-detection question “I have never lied”, leading participants to lean toward deception, which undermines its credibility in measuring the intended construct.

Psy-C1 could be removed due to its lack of clarity. Psy-C1, which states “I felt bad after playing video games for a long time”, was adapted from the survey item by [24] ([24]). However, the phrase “for a long time” poses a problem of ambiguity, as it lacks a clear definition of what constitutes “a long time”.

Findings show that “I lost concentration on other tasks because I was preoccupied with video game playing” (DD2), “I played video games when I should be sleeping” (DD4), and “I lost sleep due to the time I spent on video games” (Phy-C2) were removed due to low factor loadings. Low factor loadings indicate that an item does not strongly correlate with the underlying factor it is intended to measure. We hypothesize that the removal of these three items is due to the characteristics of the study’s participants. The majority were on-campus students, as the data were collected from middle school and university campuses. These students primarily focus on their studies and therefore have relatively few “other tasks” to attend to; additionally, most participants reside on campus, where there are regulations enforcing lights-out at designated sleeping times, thereby limiting the opportunity to play video games when they should be sleeping.

Factor misassignment led to the removal of “I was so immersed in playing video games that I forgot to eat” (DD5), which was assigned to the factor clustering with interpersonal relationship estrangement (IRE) items. The specific reasons require further exploration. Another item removed for factor misassignment was “I had skipped school so that I could play video games” (SD6), which was assigned to the factor clustering with psychological consequences (Psy-C) items. In other words, while our initial assumption was that skipping school belonged to a cluster of factors related to schooling disturbances, the data analysis showed that it was more appropriately classified as a psychological consequence. Skipping school, a significant problematic behavior, is frequently linked to an individual’s attempt to escape from stress, loneliness, or other psychological difficulties. Consequently, its assignment to the psychological consequences construct is a logical and understandable classification.

Finally, DD1 and DD3 were removed from the survey because the deletion of the other four items left only these two to measure the DD construct, which was not enough to ensure reliability.

It is noteworthy that the Physical Consequence construct explained the majority of the total variance, suggesting that physical changes may serve as a critical basis for identifying problematic gaming behaviors among adolescents. A growing body of research has demonstrated significant differences in physical characteristics between individuals with gaming addiction and those without, such as differences in bone mineral density and brain structure ([36]; [45]; [50]). Future studies could consider validating these findings in adolescents with problematic gaming behaviors.

### 5.2. Discussion on the Validated PGS-Adolescent Scale

#### 5.2.1. Strengthening the Research Focus in Developing the PGS-Adolescent Scale

One characteristic of this scale is its emphasis on measuring the negative consequences of problematic video game playing, rather than just indicators. As a result, popular constructs that are not related to negative consequences, such as tolerance and escape, were removed from this study.

Removing tolerance and other popular constructs from the scale might be controversial. However, we are not the first to do this: one manifestation of tolerance is the continually increasing gaming time, but gaming time has been removed in several studies. Specifically, [41] ([41]) did not include gaming time in the diagnostic assessment of video game dependency, stating that gaming time and similar elements “can be expected to be indicators rather than risk factors” (p. 271). Similarly, [24] ([24]) stated that “time spent on games should not be used as a basis for measuring pathological behavior”, as it is “not consistently correlated to the psychosocial concurrent validity measures”.

Likewise, neglecting social relationships tends to be a specific behavioral tendency rather than necessarily leading to negative consequences. Therefore, instead of using the exact wording when drafting survey items, we specified the negative consequences of neglecting social relationships, such as “I was not able to keep appointments due to excessive video game playing” (IRE3) and “I had been spending less time with friends or family in order to play video games” (IRE4).

However, in the schooling disturbance construct, we retained some items that do not have clearly defined negative consequences. For instance, “I neglected my studies because of my playing video games” (SD2), which was adapted from “How often do you neglect your studies, work or other important duties because of your gaming?” in the Problematic Online Gaming Questionnaire ([2]). This decision was made because our target group consists of adolescents in critical schooling periods, where learning is often not self-paced but follows a structured curriculum. Unlike maintaining social relationships, where neglect might be remedied later, neglecting studies can lead to falling behind due to the continuous progression of coursework. Given this characteristic, we believe that items such as “I neglected my studies because of playing video games” are justified for inclusion.

Another potentially controversial issue is the removal of the daily-life disturbance construct. However, upon closer examination, it becomes evident that the daily-life disturbance construct overlaps considerably with others. The definition states that daily-life disturbance “includes missing planned work, having a hard time concentrating in class or while working, suffering from lightheadedness or blurred vision, pain in the wrists or at the back of the neck, and sleeping disturbance” ([22]). Among these, “having a hard time concentrating in class or while working” can be categorized under the schooling disturbance construct, while “pain in the wrists or at the back of the neck, and sleeping disturbance” can be categorized under the physical consequences construct. The overlap makes the removal of the daily-life disturbance construct very reasonable and serves as a reminder that in scale development, it is crucial to evaluate each construct meticulously rather than mechanically combining them.

#### 5.2.2. Reducing the Number of Constructs in the PGS-Adolescent Scale

One feature of the PGS-Adolescent scale is that it reduces the number of constructs in the scale. Previous related scales tend to have a relatively high number of constructs. Specifically, the Game Addiction Scale includes seven constructs ([24]), the Problem Video Game Playing Scale has eight constructs ([47]), and the Internet Gaming Disorder Scale features nine constructs ([23]). Having too many constructs in a scale can dilute its focus, potentially impacting its validity and reliability. Additionally, it complicates the scale, making it harder for both respondents and researchers to understand and use it effectively.

In the PGS-Adolescent scale, the number of constructs was five (unvalidated version) and four (validated version), significantly fewer than the scales mentioned above. During the construct deduction process, we did not start from including a limited number of constructs; instead, we began with a large number and systematically reduced them by merging similar constructs and eliminating those unrelated to the research scope. A key criterion was to retain constructs that reflect the negative consequences of problematic behaviors. Constructs unrelated to negative consequences were removed or merged, regardless of their prominence as indicators. As [3] ([3]) emphasized, a scale should not attempt to cover all related issues but should focus on specific aspects.

#### 5.2.3. Increasing Number of Items for Each Construct in the PGS-Adolescent Scale

Another feature of the PGS-Adolescent scale is the increased number of survey items for each construct. Previous related scales have shown the problem of an insufficient number of items per construct. For example, the PVP scale, designed to measure problematic video game playing in adolescents, includes eight constructs but only has nine items in total ([47]). Similarly, the Video Game Dependency Scale has only two survey items for each of its three constructs (loss of control, withdrawal symptoms, and tolerance) ([41]).

Increasing the number of survey items for each construct ensures comprehensive coverage and measurement, minimizes errors and biases, allows for a more nuanced understanding of behaviors and symptoms, and facilitates in-depth data analysis, leading to clearer insights ([3]). Some scales place significant emphasis on this issue, such as the Smartphone Addiction Scale, which is a six-construct scale containing 48 survey items ([22]). Similarly, we need a scale with sufficient survey items for each construct to adequately measure problematic gaming in adolescents.

Notably, certain data analysis methods, such as structural equation modeling (SEM), require a minimum of three survey items per construct. This ensures that items that do not strongly contribute to the construct can be removed without compromising the model’s integrity. The PGS-Adolescent scale, with five to six items per construct, is well-suited for this purpose. Future research can use the PGS-Adolescent scale to validate the relationships between problematic gaming among adolescents and various factors, such as deviant peers, parental neglect, parental phubbing, parental knowledge, sensation seeking, conscientiousness, loneliness, escapism, and self-esteem ([1]; [19]; [25]; [17]; [44]; [48]; [27]; [37]; [54]).

Overall, the 20-item PGS-Adolescent scale consists of four constructs, which primarily detect the negative consequences brought about by problematic video game playing. A key feature of the PGS-Adolescent scale is the reduction in the number of constructs, while increasing the number of survey items for each construct. Additionally, the PGS-Adolescent scale places significant emphasis on two characteristics of this study: the need to consider the specific traits of adolescents and the focus on problematic gaming, which is a less severe condition compared to gaming addiction or gaming disorder.

## 6. Conclusions

This study developed and validated a scale named the Adolescent Problematic Gaming Scale (PGS-Adolescent), which is designed to measure problematic gaming among adolescents. After reviewing and organizing the constructs from previous related scales, we identified five constructs for the PGS-Adolescent scale. Following the scale evaluation techniques proposed by [3] ([3]) and [33] ([33]), we removed the daily-life disturbance (DD) construct, resulting in a 20-item survey that now encompasses four constructs: interpersonal relationship estrangement (IRE), schooling disturbance (SD), physical consequences (Phy-C), and psychological consequences (Psy-C).

The validated PGS-Adolescent scale significantly enhances the assessment of adolescent video game playing behaviors, offering a reliable tool for early detection and intervention. Its main benefits lie in guiding personalized prevention programs and informing policy development to promote healthy gaming practices. Furthermore, the scale provides a foundation for cross-cultural and comparative studies, contributing to a broader understanding of media-related behaviors.

One limitation of the current study is that the PGS-Adolescent scale, although in English, has not been validated among English-speaking adolescents. Although we utilized the back-translation method and most of the participants had a good command of English, the participants were primarily Chinese. Future studies could employ English-speaking adolescents to further validate the PGS-Adolescent scale. Additionally, future research might consider verifying the applicability of the theoretical framework or the scale in the current study in other contexts by replacing the key term “problematic video game playing” with terms such as “problematic social media use”, “problematic Internet use”, and “problematic short video watching”. More clinically oriented research is also essential ([47]). To better understand how problematic gaming occurs, structural characteristics of video games should also be investigated, such as in-game sound effects, the use of humor in-game, winning and losing points in-game, and leveling up ([12]; [40]; [52]).

## Figures and Tables

**Figure 1 behavsci-15-00013-f001:**
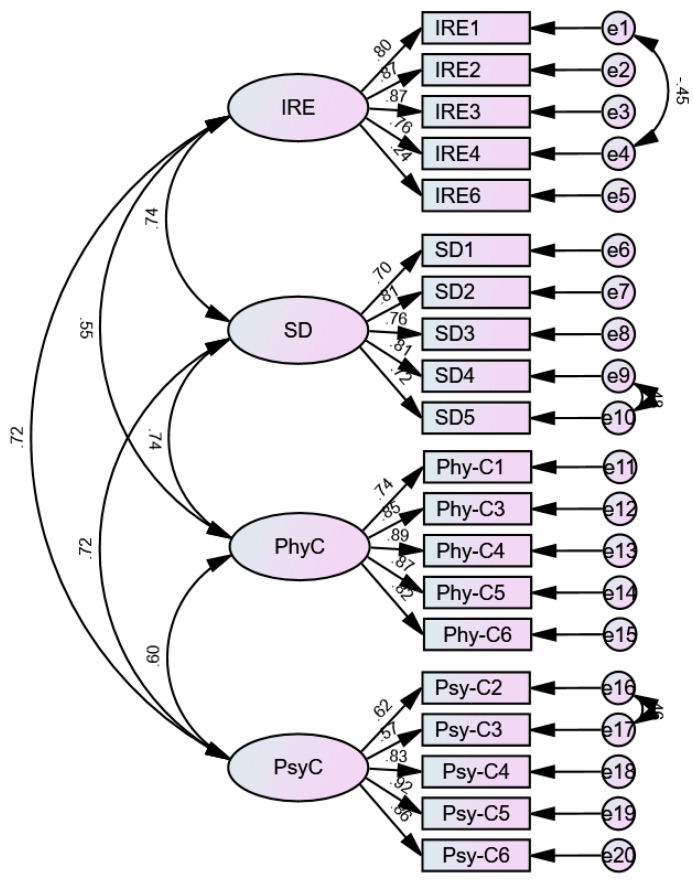
The results of the confirmatory factor analysis.

**Table 1 behavsci-15-00013-t001:** Nine related scales.

#	Scale	Abbreviation	Reference
1	Internet Gaming Disorder Scale	IGDS	([23])
2	Problematic Online Gaming Questionnaire	POGQ	([2])
3	Gaming Addiction Scale	GAS	([24])
4	Problem Video Game Playing Scale	PVP	([47])
5	Video Game Dependency Scale	VGDS	([41])
6	Smartphone Addiction Scale	SAS	([22])
7	Mobile Phone Problem Use Scale	MPPUS	([1])
8	Mobile Phone Addiction Index	MPAI	([25])
9	Mobile Phone Dependence Questionnaire	MPDQ	([5])

**Table 2 behavsci-15-00013-t002:** Constructs and similar concepts in the nine related scales.

#	Construct	Similar Concepts	Sources	Frequency (9 in Total)
1	Tolerance	Tolerance, overuse, persistence, and loss of control	IGDS, POGQ, GAS, PVP, VGDS, SAS, MPPUS, MPAI, and MPDQ	9
2	Withdrawal	Withdrawal, withdrawal symptoms, craving and withdrawal, and withdrawal/escape	POGQ, GAS, PVP, VGDS, SAS, MPPUS, MPAI, and MPDQ	8
3	Salience	Salience, preoccupation, compulsive use, and displacement	IGDS, POGQ, GAS, PVP, and VGDS	5
4	Conflict	Conflict, interpersonal conflicts, social isolation, cyberspace-oriented relationships, virtual life orientation, lies, and deception	IGDS, POGQ, GAS, PVP, VGDS, SAS, and MPDQ	7
5	Escape	Escape and positive anticipation	IGDS, PVP, SAS, MPPUS, and MPAI	5
6	Daily-life disturbance	Daily-life disturbance, negative life consequences in the areas of social, familial, work, and financial	SAS and MPPUS	2
7	Health	Disregard for physical or psychological consequences, feeling anxious and lost, and physical symptoms	PVP, MPAI, and MPDQ	3
8	Schooling disruption	Schooling disruption, productivity loss, and disturbance of concentration in class	PVP, MPAI, and MPDQ	3
9	Problems	N/A (not applicable).	IGDS and GAS	2

**Table 3 behavsci-15-00013-t003:** The initial Adolescent Problematic Gaming Scale (PGS-Adolescent).

Construct	Code	Survey Item
Daily-life disturbance (DD)	*DD1 **	*Spending a lot of time on a video game had become a habit.*
	*DD2 **	*I lost concentration on other tasks because I was preoccupied with video game playing.*
	*DD3 **	*I played video games while walking.*
	*DD4 **	*I played video games when I should be sleeping.*
	*DD5 **	*I was so immersed in video games that I forgot to eat.*
	*DD6 **	*I didn’t mind spending money on video games.*
Interpersonal relationship estrangement (IRE)	IRE1	I chose to play video games over going out with someone.
	IRE2	I had been spending less time with friends or family in order to play video games.
	IRE3	I was not able to keep appointments due to excessive video game playing.
	IRE4	I failed to meet up with a friend because I was playing video games.
	*IRE5 **	*I had lied to my parents about the time I spent playing video games.*
	IRE6	The people around me complained that I was playing video games too much.
Schooling disturbance (SD)	SD1	There had been periods when I was constantly thinking about a video game while at school.
	SD2	I neglected my studies because of playing video games.
	SD3	I played video games while studying in class, except when it was necessary.
	SD4	My productivity had decreased as a direct result of the time I spent on a video game.
	SD5	My school achievement suffered under my video gaming habits.
	*SD6 **	*I had skipped school so that I could play video games.*
Physical consequences (Phy-C)	Phy-C1	I lacked adequate sleep due to video game playing.
	*Phy-C2 **	*I lost sleep due to the time I spent on video games.*
	Phy-C3	I experienced lightheadedness or blurred vision due to video game playing.
	Phy-C4	My shoulders were stiff due to video game playing.
	Phy-C5	I felt pain in the wrists or back of the neck when playing video games.
	Phy-C6	I had headaches due to video game playing.
Psychological consequences (Psy-C)	*Psy-C1 **	*I felt bad after playing video games for a long time.*
	Psy-C2	I got anxious when I couldn’t play video games as much as I wanted.
	Psy-C3	I got irritated when bothered while playing video games.
	Psy-C4	I felt depressed when not playing video games.
	Psy-C5	I got upset when I couldn’t play video games.
	Psy-C6	I got irritable if I was unable to play video games for a few days.

Note: Items marked with an asterisk (*) have been removed from the final scale; participants were asked to complete the survey based on their experiences over the past six months.

**Table 4 behavsci-15-00013-t004:** Number of participants for each gender.

Gender	Number of Participants (Total: 448)		
	EFA	CFA	EFA and CFA
Female	179	154	333
Male	46	69	115

**Table 5 behavsci-15-00013-t005:** The removed survey items.

#	Code	Survey Item	Reason for Item-Removal
1	DD1	Spending a lot of time on a video game had become a habit.	Insufficient item retention
2	DD2	I lost concentration on other tasks because I was preoccupied with video game playing.	Low factor loadings
3	DD3	I played video games while walking.	Insufficient item retention
4	DD4	I played video games when I should be sleeping.	Low factor loadings
5	DD5	I was so immersed in video games that I forgot to eat.	Factor misassignment (to IRE)
6	DD6	I didn’t mind spending money on video games.	Low communalities
7	IRE5	I had lied to my parents about the time I spent on playing video games.	Low communalities
8	SD6	I had skipped school so that I could play video games.	Factor misassignment (to Psy-C)
9	Phy-C2	I lost sleep due to the time I spent on video games.	Low factor loadings
10	Psy-C1	I felt bad after playing video games for a long time.	Low communalities

**Table 6 behavsci-15-00013-t006:** Descriptive and factor analysis results.

Item	Mean	SD	Communalities (Extraction)	Factor/Factor Loadings			
				1	2	3	4
IRE1	2.09	1.184	0.661			0.773	
IRE2	1.82	1.013	0.788			0.904	
IRE3	1.67	0.987	0.857			0.921	
IRE4	1.49	0.869	0.656			0.734	
IRE6	1.97	1.146	0.616			0.582	
SD1	2.50	1.301	0.690		0.757		
SD2	2.44	1.278	0.782		0.816		
SD3	2.17	1.265	0.691		0.941		
SD4	2.04	1.183	0.812		0.837		
SD5	2.03	1.196	0.733		0.814		
Phy-C1	2.36	1.268	0.682	0.581			
Phy-C3	2.28	1.225	0.787	0.866			
Phy-C4	2.34	1.252	0.832	0.891			
Phy-C5	2.39	1.251	0.796	0.831			
Phy-C6	2.15	1.216	0.737	0.800			
Psy-C2	2.28	1.216	0.745				0.790
Psy-C3	2.38	1.233	0.682				0.673
Psy-C4	1.84	1.088	0.783				0.841
Psy-C5	1.82	1.069	0.784				0.723
Psy-C6	1.62	1.015	0.754				0.758
Cronbach’s alpha: 0.953				0.922	0.907	0.879	0.902
Composite reliability (CR)				0.796	0.797	0.794	0.781
Average variance extracted (AVE)				0.643	0.643	0.628	0.577

**Table 7 behavsci-15-00013-t007:** AVE, the square root of AVE and the correlations between constructs.

#	Construct	AVE	Square Root of AVE	Correlation			
				IRE	SD	Phy-C	Psy-C
1	IRE	0.628	0.792	1	0.630 **	0.599 **	0.617 **
2	SD	0.643	0.803	0.630 **	1	0.672 **	0.630 **
3	Phy-C	0.643	0.803	0.599 **	0.672 **	1	0.697 **
4	Psy-C	0.577	0.759	0.617 **	0.630 **	0.697 **	1

Note: **: correlation is significant at the 0.01 level (2-tailed).

**Table 8 behavsci-15-00013-t008:** Model fit.

Indicator	Benchmark	Result	Interpretation
Chi-square/df	Between 1 and 3	2.252	Excellent
IFI	Larger than 0.9	0.938	Excellent
TLI	Larger than 0.9	0.927	Excellent
CFI	Larger than 0.9	0.938	Excellent
CFI	Larger than 0.9	0.938	Excellent
RMSEA	Smaller than 0.08	0.075	Good

## Data Availability

The data generated and/or analyzed during the current study are available in the Figshare repository. Link: https://doi.org/10.6084/m9.figshare.26368792.v1.

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
