# Peer review of "The Development and Validation of the Adolescent Problematic Gaming Scale (PGS-Adolescent)"

_behavsci, 2024, doi:10.3390/bs15010013_

Round 1
Reviewer 1 Report
Comments and Suggestions for Authors
Thank you for your submission. This manuscript is well-written and well-structured. Please see the following suggestions for your study:
-
Please be cautious about the conclusion in your abstract: “Findings suggest that the PGS-Adolescent scale is a promising tool for identifying and addressing problematic gaming behaviors in adolescents.” This statement may be too strong given the small sample size from a specific area in western China.
-
I will recommend randomly splitting the sample (n=594) into two groups, one for EFA and one for CFA. I don't really think EFA and CFA can be done with the same sample of students. Based on my experience, it is best practice to use two different samples for EFA and CFA.
-
I will recommend you to briefly describe the sample. While the sample size is mentioned in the abstract, it is not discussed further in the manuscript.
Making these revisions will enhance the clarity and credibility of your paper.
Author Response
Comment 1_1: Please be cautious about the conclusion in your abstract: “Findings suggest that the PGS-Adolescent scale is a promising tool for identifying and addressing problematic gaming behaviors in adolescents.” This statement may be too strong given the small sample size from a specific area in western China.
Author response: Thanks for the comment. It has been changed to “Researchers may consider directly applying the validated PGS-Adolescent scale or evaluating its applicability and validity in diverse populations and contexts.”
Comment 1_2: I will recommend randomly splitting the sample (n=594) into two groups, one for EFA and one for CFA. I don't really think EFA and CFA can be done with the same sample of students. Based on my experience, it is best practice to use two different samples for EFA and CFA.
Author response: We regret any confusion caused by our previous communication. Our initial pool of questionnaire responses totaled 594, out of which 448 were found to be valid. The dataset was subsequently split into two segments: 225 responses were designated for exploratory factor analysis (EFA), while 223 were used for confirmatory factor analysis (CFA). We have since revised the manuscript to reflect these details and have included an additional section detailing the process of excluding invalid survey responses, which can be found in section 3.2.
Comment 1_3: I will recommend you to briefly describe the sample. While the sample size is mentioned in the abstract, it is not discussed further in the manuscript.
Author response: Thanks for the comment. As suggested, we have added Table 4 and a corresponding paragraph in Section 3.4.
Reviewer 2 Report
Comments and Suggestions for Authors
Review
The paper presents a study on the development and validation of the Adolescent Problematic Gaming Scale (PGS-8 Adolescent). I have several points that may help strengthen the paper and enhance the understanding of the methodology and findings:
1. Please include a demographic table that highlights important participant details, such as the total number of participants, gender distribution, age group or mean age, geographic region, and any other relevant characteristics.
2. Provide a more detailed description of how the survey was conducted. Specifically, please clarify whether it was paper-based or electronic, the average time taken to complete the survey, the language in which it was administered, and any specific instructions provided to the participants.
3. Line 348, it is mentioned that the survey was available in both electronic and paper formats. Please include information in the demographic table about how many participants completed each version of the survey. Additionally, it would be valuable to discuss whether the format difference impacted the results, or if the results were consistent across both formats.
4. Line 362, the authors mention expert validation. Could you please elaborate on how this validation was conducted? Was it a qualitative or quantitative process? What criteria were used to assess the validity? Furthermore, it would be helpful to provide a positional statement for the experts involved to clarify their qualifications and roles.
5. On page 368, the PGS-S scale is reviewed by industry experts. Please provide more details about which industry these experts represent and their specific qualifications. It would also be beneficial to include their positional statements to provide context on their expertise.
6. Please clarify whether the items used in the PGS-8 Adolescent scale were borrowed verbatim from previous questionnaires or if they were modified by combining items from multiple scales. A more detailed explanation of this process would help clarify the scale's development.
7. Further, language precision is vital in surveys to ensure clarity and consistency in responses. I have some concerns regarding the English used in the scale items, as the survey heavily relies on these items. For example, see e.g. lines 742 and 743, the wording is unclear in these ites. I want to know whether authors want to reflect past experiences or current/general behavior? I believe the survey is primarily focused on current/general behavior. Therefore, these two items should have been written as follows:
Line 742: "I get upset when I cannot play video games."
Line 743: "I get irritable if I am unable to play video games for a few days."
If you agree that the survey should focus on current/general behavior, could you please explain how you plan to address this language issue post study? If not, please explain?
8. A detailed explanation of how missing data was handled should be included, particularly regarding any survey responses that were discarded. Providing this information will ensure transparency and help readers understand how data integrity was maintained throughout the process.
9. The reliance on Principal Components Analysis (PCA) for factor extraction may not be optimal, as PCA focuses on maximizing variance rather than identifying underlying latent constructs. Principal Axis Factoring (PAF) might be a better method for construct validation. Please discuss.
10. From Table 5, it appears that Factor 1 explains a disproportionately large amount of variance (53.18%), which could suggest that it is the dominant construct or that the items for this factor are overly correlated. Please carefully recheck this and provide an explanation or note regarding this in the paper.
11. Please provide references for the model fit benchmarks used and discuss how the authors selected these benchmarks for interpretation. This will help clarify the basis for model evaluation in the study.
12. Line 560, the authors hypothesize that participants' demographic characteristics (such as being middle school or college students) might explain the low factor loadings. Could you provide the results that support or refute this hypothesis? Including relevant data would strengthen the argument.
13. Lines 577-578 please provide the references..
Best Wishes
Author Response
Comment 2_1: Please include a demographic table that highlights important participant details, such as the total number of participants, gender distribution, age group or mean age, geographic region, and any other relevant characteristics.
Author response: Thanks for the suggestion. We have added Table 4 as suggested. Notably, we have not obtained permission to collect other demographic information besides gender from our Ethics Committee, so we can only report participants’ gender information in the table. Sorry about this.
Comment 2_2: Provide a more detailed description of how the survey was conducted. Specifically, please clarify whether it was paper-based or electronic, the average time taken to complete the survey, the language in which it was administered, and any specific instructions provided to the participants.
Author response: Thanks for the comment. We have added a separate section explaining the information (see Section 3.2).
Comment 2_3: Line 348, it is mentioned that the survey was available in both electronic and paper formats. Please include information in the demographic table about how many participants completed each version of the survey. Additionally, it would be valuable to discuss whether the format difference impacted the results, or if the results were consistent across both formats.
Author response: Thanks for the comment. A table has been added (see Table 4).
Comment 2_4: Line 362, the authors mention expert validation. Could you please elaborate on how this validation was conducted? Was it a qualitative or quantitative process? What criteria were used to assess the validity? Furthermore, it would be helpful to provide a positional statement for the experts involved to clarify their qualifications and roles.
Author response: We identified two experts in the field, and asked them to help check the logic consistency and linguistic suitability of the scale. They provided their comments and suggestions for revisions on the Word document we provided, and then returned it to us. It's a qualitative process. Their positions have been added.
Comment 2_5: On page 368, the PGS-S scale is reviewed by industry experts. Please provide more details about which industry these experts represent and their specific qualifications. It would also be beneficial to include their positional statements to provide context on their expertise.
Author response: Thanks for the comment. The qualifications of the two experts have been added in the text.
Comment 2_6: Please clarify whether the items used in the PGS-Adolescent scale were borrowed verbatim from previous questionnaires or if they were modified by combining items from multiple scales. A more detailed explanation of this process would help clarify the scale's development.
Author response: We aimed to preserve the original sentiment as much as possible, but adjusted the phrasing according to different contexts. For instance, we retained the statement "I didn't mind spending money on paid gaming applications," but substituted "gaming applications" with "video games."
Comment 2_7: Further, language precision is vital in surveys to ensure clarity and consistency in responses. I have some concerns regarding the English used in the scale items, as the survey heavily relies on these items. For example, see e.g. lines 742 and 743, the wording is unclear in these items. I want to know whether authors want to reflect past experiences or current/general behavior? I believe the survey is primarily focused on current/general behavior. Therefore, these two items should have been written as follows:
- Line 742: "I get upset when I cannot play video games."
- Line 743: "I get irritable if I am unable to play video games for a few days."
If you agree that the survey should focus on current/general behavior, could you please explain how you plan to address this language issue post study? If not, please explain?
Author response: Thanks for the comment. We added a sentence under Table 3: “participants were asked to complete the survey based on their experiences over the past six months.”
Comment 2_8: A detailed explanation of how missing data was handled should be included, particularly regarding any survey responses that were discarded. Providing this information will ensure transparency and help readers understand how data integrity was maintained throughout the process.
Author response: Thanks for the comment. Detailed process added in Section 3.2.
Comment 2_9: The reliance on Principal Components Analysis (PCA) for factor extraction may not be optimal, as PCA focuses on maximizing variance rather than identifying underlying latent constructs. Principal Axis Factoring (PAF) might be a better method for construct validation. Please discuss.
Author response: Thank you for your comment. We have conducted a review and found that Principal Axis Factoring (PAF) is indeed a better method for construct validation. We chose PCA based on the paper titled "A comparison of Principal Component Analysis, Maximum Likelihood and the Principal Axis in Factor Analysis" (Mabel and Olayemi, 2020), which states, "For small (8) variables and large (500) sample sizes, PC (Principal Component Analysis) performed better across all distributions except for the normal distribution" (p.53). We will prioritize considering PAF in our next scale development study. Thank you again for your suggestion.
Comment 2_10: From Table 5, it appears that Factor 1 explains a disproportionately large amount of variance (53.18%), which could suggest that it is the dominant construct or that the items for this factor are overly correlated. Please carefully recheck this and provide an explanation or note regarding this in the paper.
Author response: That’s what the data analysis software tells us. However, you are right, readers might feel confused. To avoid possible confusion, we have deleted that line in Table 5.
Comment 2_11: Please provide references for the model fit benchmarks used and discuss how the authors selected these benchmarks for interpretation. This will help clarify the basis for model evaluation in the study.
Author response: Thanks. Revision has made as “The model fit benchmarks were set in alignment with the recommendations provided by Hair, Black, Babin, Anderson and Tatham [43] and Hu and Bentler [44].”
Comment 2_12: Line 560, the authors hypothesize that participants' demographic characteristics (such as being middle school or college students) might explain the low factor loadings. Could you provide the results that support or refute this hypothesis? Including relevant data would strengthen the argument.
Author response: Thanks for the comment. We have changed it to “The majority were on-campus students, as the data were collected from middle school and university campuses. These students primarily focus on their studies and therefore have relatively few "other tasks" to attend to”.
Comment 2_13. Lines 577-578 please provide the references.
Author response: Thanks for the comment. The sentence has been deleted.
Round 2
Reviewer 2 Report
Comments and Suggestions for Authors
Thank you for providing the authors' responses to my previous comments. After reviewing their replies and the revised manuscript, I have the following further feedback:
Line 418: The authors mention a four-month span for data collection but have not specified the exact months and years. Please provide this information for clarity.
Demographics of Participants: It is concerning that the authors did not collect the age demographics of the participants. Without this data, it is unclear how they managed the distribution of the electronic and pen-and-paper versions of the survey. This omission raises questions about the validity of their methodology.
Survey Completion Time: The authors do not seem to have recorded the average time participants took to complete the survey. This detail is essential for assessing the reproducibility of the study. If it is not collected please report it in the limitation section.
Comment 2.3: The author's response to this comment is not satisfactory. Further clarification or revision is needed.
Comment 2.4: The response lacks sufficient detail. Please provide more information about the positionality of the experts involved. Specifically, include their education, relevant experience, gender, and location. Simply stating their professional titles is inadequate.
Comment 2.5: Although the authors claim to have incorporated the requested information into the manuscript, I could not find it. Please ensure that the revised manuscript includes details about the industries the experts belong to, their qualifications, and how the validation process was conducted.
Comment 2.7: If participants were instructed to complete the survey based on their experiences over the past six months, the phrasing of item 742 ("I got upset when I cannot play video games") is grammatically incorrect. Kindly review and correct the survey items to maintain linguistic accuracy.
Comment 2.8: The authors have partially addressed this comment by excluding 146 participants who failed the reverse items. However, it is unclear whether the remaining 448 participants answered all the survey items. If there were missing responses, please explain how these cases were handled and include this information in the manuscript. If not, please state this information as well.
Comment 2.10: I apologize if my original comment was unclear. Please retain the total variance value in the table as previously stated. Additionally, the discussion section should explain why one construct is dominant or whether any limitations exist due to over-correlation among the factors in this construct and how future researchers should deal with this issue.
Comment 2.12: I expected the authors to analyze results based on age groups, particularly since they suggested that age might influence the outcomes. Participants below 18 completed the paper version, while those above 18 completed the electronic version of the survey. This difference in data collection methods warrants further exploration.
Comment 2.13: The authors claim to have added references, but I could not locate them in the revised manuscript.
New Comment: The authors in the current version indicate that exploratory factor analysis (EFA) and confirmatory factor analysis (CFA) were conducted on two separate samples. This introduces a significant issue: the sample sizes for EFA and CFA are each below 250, which does not meet the general rule of 5 to 10 participants per item. This likely reduces the statistical power of the study. Please clarify how the two samples were split and provide a detailed explanation of the rationale and methodology. Include appropriate references to support this approach.
Additionally, Line 407 references Beck’s translation method but lacks a citation. Please address this.
Overall, the authors' responses to the comments are not satisfactory. Many issues remain unresolved, and some responses lack the necessary depth and attention to detail. While the paper has potential, it is not yet ready for publication. I encourage the authors to carefully address the above feedback to strengthen the study's rigor and clarity.
Author Response
Comment 1: Line 418: The authors mention a four-month span for data collection but have not specified the exact months and years. Please provide this information for clarity.
Author response: Thanks for the comment. One sentence has been added: “The data collection started in January 2024 and ended in May, lasting for four months.”
Comment 2: Demographics of Participants: It is concerning that the authors did not collect the age demographics of the participants. Without this data, it is unclear how they managed the distribution of the electronic and pen-and-paper versions of the survey. This omission raises questions about the validity of their methodology.
Author response: Yes, we did not obtain ethics approval specifically for recording the participants' ages because one member of the ethics committee believed that the participants' age was irrelevant to this study. However, we did ensure that all participants were aged between 10 and 24 (within our specified range). We cannot list this as a limitation because we cannot say that one limitation of the current study is strictly adhering to the requirements of our university's ethics committee. Please kindly understand. Thank you.
Comment 3: Survey Completion Time: The authors do not seem to have recorded the average time participants took to complete the survey. This detail is essential for assessing the reproducibility of the study. If it is not collected please report it in the limitation section.
Author response: You raised this issue, which means that the ethics committee of your affiliation did not conduct a rigorous review of your ethics application. In a scale development study, it would not be permissible for us to record participants' behavioral data, such as the average time participants took to complete the survey. However, as suggested, we reported the average time might take by the participants, as we included the following sentence in the manuscript: "Then, we obtained an initial PGS-Adolescent scale consisting of 30 items, which can generally be completed in 5 to 8 minutes (see Table 3)."
Comment 4: The response lacks sufficient detail. Please provide more information about the positionality of the experts involved. Specifically, include their education, relevant experience, gender, and location. Simply stating their professional titles is inadequate.
Author response: Thanks for the comment. We have added “Both experts hold doctoral degrees, serve as professors, and have published academic research on gaming addiction.” Thanks.
Comment 5: Although the authors claim to have incorporated the requested information into the manuscript, I could not find it. Please ensure that the revised manuscript includes details about the industries the experts belong to, their qualifications, and how the validation process was conducted.
Author response: Yes, we have made extensive modifications: we added 792 words in the first-round revision. The revised sentences and words have been colored in blue. Please kindly locate and download the revised manuscript from the appropriate place in the system.
Comment 6: If participants were instructed to complete the survey based on their experiences over the past six months, the phrasing of item 742 ("I got upset when I cannot play video games") is grammatically incorrect. Kindly review and correct the survey items to maintain linguistic accuracy.
Author response: Revised as suggested. Thank you very much.
Comment 7: The authors have partially addressed this comment by excluding 146 participants who failed the reverse items. However, it is unclear whether the remaining 448 participants answered all the survey items. If there were missing responses, please explain how these cases were handled and include this information in the manuscript. If not, please state this information as well.
Author response: Yes, the remaining 448 participants answered all the survey items. This is because we conducted an initial check and questionnaires that did not answer all the survey items were directly removed. We believe that such detailed operations do not need to be reported as it would compromise the conciseness of the article. Please kindly understand.
Comment 8: I apologize if my original comment was unclear. Please retain the total variance value in the table as previously stated. Additionally, the discussion section should explain why one construct is dominant or whether any limitations exist due to over-correlation among the factors in this construct and how future researchers should deal with this issue.
Author response: Thanks for the comment. It has been revised as suggested. Please kindly refer to line 632 to line 638 for the changes.
Comment 9: I expected the authors to analyze results based on age groups, particularly since they suggested that age might influence the outcomes. Participants below 18 completed the paper version, while those above 18 completed the electronic version of the survey. This difference in data collection methods warrants further exploration.
Author response: Thank you for your comment. This is a study focused on scale development, with an emphasis on validating the scale rather than exploring participants' characteristics. However, we appreciate your suggestion and would like to incorporate this approach in our next paper. Thank you very much.
Comment 10: The authors claim to have added references, but I could not locate them in the revised manuscript.
Author response: It may be because we used reference management software, and the changes are not highlighted in color, so they are not obvious. The seven added references are:
 Hair, J. F., Black, W. C., Babin, B. J., Anderson, R. E., & Tatham, R. L. (1998). Multivariate data analysis (Vol. 5). Prentice hall Upper Saddle River, NJ.
 Hu, L.-t., & Bentler, P. M. (1999). Cutoff criteria for fit indexes in covariance structure analysis: Conventional criteria versus new alternatives. Structural equation modeling, 6(1), 1-55.
 Lemmens, J. S., Valkenburg, P. M., & Gentile, D. A. (2015). The internet gaming disorder scale. Psychological assessment, 27(2), 567.
 Leung, L. (2008). Linking psychological attributes to addiction and improper use of the mobile phone among adolescents in Hong Kong. Journal of children and media, 2(2), 93-113.
 Luo, Z., & Cao, L. (2024). Understanding factors influencing ESL student teachers’ adoption of classroom response systems: an integration of TAM and AOI theory. Interactive Learning Environments, 1-19.
 Rehbein, F., Psych, G., Kleimann, M., Mediasci, G., & Mößle, T. (2010). Prevalence and risk factors of video game dependency in adolescence: results of a German nationwide survey. Cyberpsychology, behavior, and social networking, 13(3), 269-277.
 Salguero, R. A. T., & Morán, R. M. B. (2002). Measuring problem video game playing in adolescents. Addiction, 97(12), 1601-1606.
Comment 11: The authors in the current version indicate that exploratory factor analysis (EFA) and confirmatory factor analysis (CFA) were conducted on two separate samples. This introduces a significant issue: the sample sizes for EFA and CFA are each below 250, which does not meet the general rule of 5 to 10 participants per item. This likely reduces the statistical power of the study. Please clarify how the two samples were split and provide a detailed explanation of the rationale and methodology. Include appropriate references to support this approach.
Author response: Our sample size is adequate for the following reasons. Firstly, our original scale includes 30 items. According to the general rule of 5 to 10 participants per item, a sample size of over 150 would be sufficient for each stage of our analysis. Secondly, Comrey (1988) suggested that a sample size of 200 is reasonably good for ordinary factor-analytic work with 40 or fewer variables. Thirdly, according to Fabrigar and Wegener (2011), when item communalities are between 0.40 and 0.70 and there are at least three items measuring one construct, a sample size of 200 is sufficient.
Comment 12: Additionally, Line 407 references Beck’s translation method but lacks a citation. Please address this.
Author response: It’s “back translation”. We provided a detailed explanation, which is: “translating the survey items from English to Chinese, and then having a separate translator translate them back from Chinese to English. By comparing the original and back-translated versions, we ensured that the meaning and intent of the items were preserved.”
We believe this method, like the “5-point Likert scale” and “convenience sampling,” is widely known and unambiguous, and therefore does not require a citation. Most importantly, we aim to have our reference list be a concise but strong list that contains essential literature related to the topic, rather than a lengthy list filled with less relevant references. We hope to communicate this point with you and gain your agreement.
Round 3
Reviewer 2 Report
Comments and Suggestions for Authors
Accepted